# Characterization and Homology Modeling of Catalytically Active Recombinant PhaC_Ap_ Protein from *Arthrospira platensis*

**DOI:** 10.3390/biology12050751

**Published:** 2023-05-20

**Authors:** Chanchanok Duangsri, Tiina A. Salminen, Marion Alix, Sarawan Kaewmongkol, Nattaphong Akrimajirachoote, Wanthanee Khetkorn, Sathaporn Jittapalapong, Pirkko Mäenpää, Aran Incharoensakdi, Wuttinun Raksajit

**Affiliations:** 1Program of Animal Health Technology, Faculty of Veterinary Technology, Kasetsart University, Bangkok 10900, Thailandfvetspj@ku.ac.th (S.J.); 2Structural Bioinformatics Laboratory and InFLAMES Research Flagship Center, Biochemistry, Faculty of Science and Engineering, Åbo Akademi University, 20520 Turku, Finland; 3Department of Physiology, Faculty of Veterinary Medicine, Kasetsart University, Bangkok 10900, Thailand; 4Division of Biology, Faculty of Science and Technology, Rajamangala University of Technology Thanyaburi (RMUTT), Thanyaburi, Pathumthani 12110, Thailand; 5Faculty of Technology, University of Turku, 20014 Turku, Finland; 6Laboratory of Cyanobacterial Biotechnology, Department of Biochemistry, Faculty of Science, Chulalongkorn University, Bangkok 10330, Thailand; 7Academy of Science, Royal Society of Thailand, Bangkok 10300, Thailand

**Keywords:** *Arthrospira platensis*, PhaC, PHA synthase, Polyhydroxybutyrate, 3HB-CoA

## Abstract

**Simple Summary:**

The cyanobacterium *Arthrospira platensis* contains PHA synthase Class III (PhaC_Ap_), which can produce short chain length (SCL) PHB under nitrogen-depleted conditions. In this study, we cloned a gene encoding PhaC from *A. platensis* into *Escherichia cloni* ^®^10G cells to produce rPhaC_Ap_ protein. The *V_max_*, *K_m_*, and *k_cat_* values for β-3-hydroxybutyryl coenzyme A (3HB-CoA) of the purified rPhaC_Ap_ were investigated. Size-exclusion chromatography revealed that rPhaC_Ap_ exists as an active dimer. The overall fold and catalytic triad residues were predicted using the 3D structural model for rPhaC_Ap_. These results are discussed with respect to the dimerization mechanism of PhaC_Ap_, which has not yet been clarified.

**Abstract:**

Polyhydroxybutyrate (PHB) is a biocompatible and biodegradable polymer that has the potential to replace fossil-derived polymers. The enzymes involved in the biosynthesis of PHB are β-ketothiolase (PhaA), acetoacetyl-CoA reductase (PhaB), and PHA synthase (PhaC). PhaC in *Arthrospira platensis* is the key enzyme for PHB production. In this study, the recombinant *E. cloni* ^®^10G cells harboring *A. platensis phaC* (rPhaC_Ap_) was constructed. The overexpressed and purified rPhaC_Ap_ with a predicted molecular mass of 69 kDa exhibited *V_max_*, *K_m_*, and *k_cat_* values of 24.5 ± 2 μmol/min/mg, 31.3 ± 2 µM and 412.7 ± 2 1/s, respectively. The catalytically active rPhaC_Ap_ was a homodimer. The three-dimensional structural model for the asymmetric PhaC_Ap_ homodimer was constructed based on *Chromobacterium* sp. USM2 PhaC (PhaC_Cs_). The obtained model of PhaC_Ap_ revealed that the overall fold of one monomer was in the closed, catalytically inactive conformation whereas the other monomer was in the catalytically active, open conformation. In the active conformation, the catalytic triad residues (Cys151-Asp310-His339) were involved in the binding of substrate 3HB-CoA and the CAP domain of PhaC_Ap_ involved in the dimerization.

## 1. Introduction

Energy demand worldwide is increasing continuously due to human population and economic growth. Higher consumption of fossil fuels leads to higher greenhouse gas emissions, particularly of CO_2_, which contribute to global warming. Petrochemical-based plastics, which have been used in daily life for decades, cannot be decomposed by biological processes and accumulate in the environment. The global demand for biodegradable plastic as a mandatory substitute for synthetic plastics is augmented by considering their biocompatibility, biodegradability, nontoxicity, and renewability properties [1,2]. Among the biopolymers, poly (3-hydroxybutyrate) (PHB) is the most well-characterized of the polyhydroxyalkanoates (PHAs) that are produced by a large variety of microorganisms such as bacteria, fungi, and cyanobacteria under unbalanced growth [3]. The PHB has properties similar to some traditional synthetic plastics such as polypropylene and polyethylene [4,5]. The PHB biosynthetic pathway consists of three enzymatic reactions catalyzed by three distinct enzymes encoded by the *phaABC* operon in many cyanobacterial species [6]. The first reaction begins with the condensation of two acetyl-CoA molecules into acetoacetyl-CoA by β-ketothiolase (encoded by *phaA*). The second reaction is the conversion of acetoacetyl-CoA to 3-hydroxybutyryl-CoA by acetoacetyl-CoA reductase (encoded by *phaB*). Finally, the 3-hydroxybutyryl-CoA monomers are polymerized into poly (3-hydroxybutyrate) (PHB) by PHA synthase (encoded by *phaC*) (Appendix A).

PHA synthases (PhaC) are classified into four classes based on subunit composition, primary sequences, and substrate specificity [7,8]. The structure and properties have been characterized for the Class I PhaC proteins from *Cupriavidus necator* (PhaC_Cn_) [7], *Chromobacterium* sp. USM2 (PhaC_Cs_) [9], *Rhodovulum sulfidophilum* (PhaC_Rs_) [10], and *Aquitalea* sp. USM4 (PhaC_As_) [11]. Class II PhaC is characterized from *Pseudomonas* spp. (PhaC_Ps_) [12], Class III PhaC from *Chromatium vinosum* (PhaC_Cv_), and Class IV PhaC from *Bacillus* spp. (PhaC_Bs_) [13]. PHB can be categorized into two primary groups based on the carbon atom count present in the monomer units. The first group is referred to as short-chain-length (SCL) PHB and consists of repeat units made up of hydroxy fatty acids containing carbon atoms ranging from C3 to C5. This type of PHB is generated through PhaC Class I, III, and IV. The second group is known as medium-chain-length (MCL) PHB. The repeat units in this group are hydroxy fatty acids of C6-C14 carbon atoms, and they are produced by PhaC Class II [14].

*Arthrospira platensis* is a filamentous cyanobacterium that grows naturally in alkaline salt lakes. *A. platensis* are the most employed because of their high growth rate, simple cultivation, and relatively easy harvesting [15]. *A. platensis* contains PHA synthase Class III (PhaC_Ap_), which can produce SCL-PHB. Most *Arthrospira* species produce large amounts of PHB under nitrogen-depleted conditions [16]. It is worth noting that the highest content of polyhydroxybutyrate (PHB) was found in *Arthrospira platensis* cultures that were grown under nitrogen-deprived conditions for three days, while being supplemented with 0.50% (*w*/*v*) acetate. This resulted in the maximum PHB content of 19.2% of cell dry weight [17], which is much higher than the unicellular cyanobacterium *Synechocystis* sp. PCC 6803 under optimized conditions, which showed a PHB content of 13.1% [18]. Therefore, *A. platensis* is considered to be one of the most viable candidates for industrial-scale production of PHB. To obtain a more comprehensive understanding of the function of PhaC, we created recombinant *Escherichia cloni*^®^10G cells that contained *A. platensis phaC*. The expressed *A. platensis* PhaC protein was purified and evaluated for its enzymatic activity and oligomeric structure in vitro. We also constructed homology 3D models for *A. platensis* PhaC and analyzed the active site for its ligand-binding properties.

## 2. Materials and Methods

### 2.1. Bacterial Strains and Culture Conditions

The *Arthrospira platensis* was cultivated in a 250-mL Erlenmeyer flask with 50 mL of Zarrouk medium (pH 10.0) at 32 °C under a continuous fluorescent lamp emitting 40 μmol photon/m^2^/s on a rotatory shaker at 120 rpm [14]. *Escherichia cloni^®^*10G (Lucigen) was used for cloning and expression analysis. Luria–Bertani (LB) media supplemented with 100 g/mL kanamycin was used to grow and maintain cells containing recombinant plasmids. Following overnight cell growth at 37 °C, the growth of *E. cloni*^®^10G was measured in terms of turbidity (OD_600_). *E. cloni^®^*10G strains were grown in LB medium at 37 °C and 120 rpm in an incubator. Expresso^®^ pSol-Tsf vector (Lucigen) was used as a vector for cloning.

### 2.2. Cloning of phaC_Ap_ in pSol-Tsf Plasmid

The open reading frame of *phaC_Ap_* gene (1095 bp) encoding PhaC_Ap_ protein was amplified from *A. platensis* genomic DNA using primers: Fpha: 5′-AAT CTG TAC TTC CAG GGT ATG TTA CCT-3′ and Rpha: 5′-GTG GCG GCC GCT CTA TTA TTA CTC TCG-3′ (underlined sequence indicates primers that add flanking sequence identical to those found at the ends of linear pSol-Tsf vector). PCR was performed with a temperature program starting at 94 °C for 10 min, followed by 30 cycles of 94 °C for 45 s, 52 °C for 45 s, 72 °C for 45 s, and a final elongation at 72 °C for 10 min. The *phaC_Ap_* gene, which was amplified to a length of 1131-bp, was extracted and purified using the Silica Bead DNA Gel Extraction Kit (Thermo Fisher Scientific), and the fragment was cloned into the Expresso^®^ pSol-Tsf vector (Lucigen). The recombinant plasmid (pSol-Tsf-phaC) harboring the SelecTEV^TM^ Protease, 6x-His-tagged fusion, and terminator recognition sites was transformed into *E. cloni*^®^10G, and the presence of inserts with correct sequence was verified by colony PCR and further by sequencing.

### 2.3. Expression and Purification of rPhaC_Ap_

Fresh 500 mL LB broth containing 100 μg/mL kanamycin was inoculated with transformed cells harboring recombinant (pSol-Tsf-phaC) or empty vectors (pSol-Tsf), and allowed to grow at 37 °C. When the OD_600_ of the culture reached ~0.5, cells were induced with a final concentration of 0.2% (*w*/*v*) L-rhamnose. This culture was incubated at 30 °C for 4 h with shaking at 120 rpm. Overexpressed cells were collected and subjected to three MQ water washes. The cell pellets were suspended in buffer pH 7.5 (50 mM Na_2_HPO_4_, 300 mM NaCl) and lysed by sonication (20 kHz) at 32 °C for 10 min, followed by centrifugation to remove cell debris. The supernatant containing recombinant 6x-His-tagged fusion protein (rPhaC_Ap_) was loaded onto a Ni-NTA column (Ni Sepharose^TM^ 6 Fast Flow, GE Healthcare Bio-sciences AB), then washed with five column volumes of binding buffer (50 mM Na_2_HPO_4_ pH 7.5, 300 mM NaCl, 10 mM imidazole). The rPhaC_Ap_ was eluted from the column with elution buffer (50 mM Na_2_HPO_4_, pH 7.5, 300 mM NaCl, 90 mM imidazole). The protein purity was determined by using 12% sodium dodecyl sulfate-polyacrylamide gel electrophoresis (SDS-PAGE) and Western blotting. Protein concentration was determined by using a Bio-Rad protein assay kit and BSA as standard [19].

### 2.4. In Vitro Enzymatic Assay of the Purified rPhaC_Ap_

The activity of the purified rPhaC_Ap_ was determined by estimating the amount of CoA (reaction product) using β-3-hydroxybutyryl coenzyme A (3HB-CoA). This assay was carried out in the reaction mixture (1 mL) containing 50 μM 3HB-CoA, 50 μM 5,5′-dithiobis-(2-nitrobenzoic acid) (DTNB), BSA (0.2 mg/mL) in 100 mM Tris-HCl buffer (pH 8.0) and enzyme, which was previously reported by [20]. Incubation was carried out at 30 °C for 10 min. The liberation of CoA was determined by measuring the absorbance at 412 nm with a spectrophotometer. The enzyme activity was expressed as µmol/min/mg. The measurement was carried out in triplicate. Protein concentration was determined by using a Bio-Rad protein assay kit and BSA as standard [19].

### 2.5. Determination of Steady-State Kinetic Constants of rPhaC_Ap_

The *K_m_* and *k_cat_* values were calculated using the initial velocity data obtained by varying the concentration of 3HB-CoA from 10 μM to 400 μM. The initial velocity data were fitted to Michaelis–Menten equation *V_0_* = *V_max_*[S]/(*K_m_* + [S]) (where *V_0_* is the initial velocity, *V_max_* is the maximum velocity, [S] is the substrate concentration, and *K_m_* is the Michaelis constant) with GraphPad Prism 9.5.1 software. The *K_m_* represents the equilibrium constant for dissociation of the enzyme from the substrate. The *k_cat_* value was calculated from the ratio of *V_max_* and enzyme concentration. Each reaction was initiated by the addition of 100 μg of crude enzyme to 1 mL of reaction mixture.

### 2.6. Production of Polyclonal Antibodies against rPhaC_Ap_

Two healthy, 12-week-old male New Zealand rabbits were maintained in the experimental animal facility. Purified rPhaC_Ap_ (100 μg) was emulsified with an equal volume of Freund’s Complete Adjuvant (FCA) and subcutaneously injected into rabbits for the first immunization. Three booster injections of the same protein mixed with FCA were given to each rabbit on the 14^th^, 21^st^, and 28^th^ day. The rabbits were bled via the marginal ear vein prior to the first dose and at 7-day intervals, and the serum antibody titer was measured using ELISA. The rabbit anti-rPhaC_Ap_ polyclonal antibody serum was kept for further experiment. The body weight, temperature, complete blood count, and behavior of the rabbits in the home cage, upon handling, and in an open field did not differ significantly among the immunization groups during the 7-week assessment period.

### 2.7. Western Blot Analysis

The 6x-His-tagged fusion rPhaC_Ap_ protein (69 kDa) was purified and analyzed by resolving in a 12% SDS-PAGE gel. The separated protein was then transferred to a nitrocellulose membrane (Bio-Rad) using a Trans-Blot^®^ Semi-Dry system (Bio-Rad) with an applied voltage of 10V for 60 min. The nitrocellulose membranes were incubated with 5% (*w*/*v*) non-fat powdered milk in PBST buffer (25 mM phosphate-buffer pH 7.4, 150 mM NaCl, 0.1% (*v*/*v*) Tween 20) at 4 °C overnight. The membranes were washed three times by PBST buffer. Subsequently, the nitrocellulose membranes were incubated with either a 1:5000 dilution of rabbit anti-His-tagged polyclonal antibody (Thermo Fisher Scientific) or a 1:10,000 dilution of rabbit anti-rPhaC_Ap_ polyclonal antibody in PBST buffer for 3 h at room temperature. Next, the membranes were washed three times with PBST buffer and then incubated with a 1:2000 dilution of HRP-conjugated goat anti-rabbit IgG (Sigma-Aldrich) in PBST buffer containing 2% non-fat powdered milk for 90 min at room temperature. Following the washing step, the nitrocellulose membranes were treated with diaminobenzidine (DAB) using a DAB substrate kit (Thermo Fisher Scientific) at room temperature for 5–10 min. A brown band on a nitrocellulose membrane was observed, indicating the presence of the rPhaC_Ap_ protein.

### 2.8. Size-Exclusion Chromatography

The reaction mixture consisting of purified rPhaC_Ap_ and 100 mM sodium phosphate (pH 7.4) was incubated for 10 min. The reaction mixtures were then loaded onto a Sephadex-G50 column (Merck, Darmstadt, Germany) equilibrated with 100 mM sodium phosphate (pH 7.4). The reaction mixture containing rPhaC_Ap_ was eluted with the same buffer at a flow rate of 1 mL/min. The eluted samples were measured for absorbance at 280 nm, and their retention times were compared to those of molecular standards. The standards used for comparison were alcohol dehydrogenase (150 kDa, 18.5 min), BSA (66 kDa, 24.5 min), and ovalbumin (44 kDa, 27.5 min).

### 2.9. Sequence and Multiple Sequence Alignment Analysis

The amino acid sequence of PhaC_Ap_ was retrieved from UniProtKB (UniProt code D4ZNW6). BLAST (Basic Local Alignment Search Tool) was used to identify the structural templates for modeling PhaC_Ap_. The Protein Data Bank (PDB, http://www.rcsb.org (accessed on 11 May 2020); Berman et al., 2000), which is available at http://www.rcsb.org (accessed on 13 May 2020) and contains a collection of protein structures, was used as the target database. The BLAST searches revealed 4 PhaC structures with expectation values (e-value) below 1 × 10^−17^: two structures from *Cupriavidus necator* H16 (*Ralstonia eutropha*) (PDB codes 5T6O and 5HZ2), and *Chromobacterium* sp. USM2 (PDB codes:5XAV and 6K3C). Next, we created a structure-based sequence alignment of the PhaC structures from *Cupriavidus necator* H16 (5HZ2) and from *Chromobacterium* sp. USM2 (6K3C) by superimposing the structures with VERTAA [21]. Since the sequence identity of these structures to PhaC_Ap_ was only around 20%, we searched for more sequences similar to PhaC_Ap_ and aligned the *Synechocystis* sp. PCC 6803 PhaC ((PhaC_Ss_; WP_041425936.1), *Bernardetia litoralis* PhaC (PhaC_Bl_; WP_014797661.1), and *Aromatoleum buckelii* PhaC (PhaC_Ab_; WP_169197587.1) to the pre-aligned structure-based alignment using MALIGN in Bodil [22]. The secondary structure for the PhaC_Ap_ sequence was predicted using Jpred [23]. The final sequence alignment figure was prepared with ESPript 3.0 [24].

### 2.10. Homology Modelling

The 3D model of PhaC_Ap_ was created using the multiple sequence alignment and the crystal structure of PhaC_Cs_ from *Chromobacterium* sp. USM2 (PDB code: 6K3C) as a template, with a resolution of 1.8 Å. MODELLER generated ten models for PhaC_Ap_, and the one with the lowest energy, as determined by the MODELLER objective function, was selected for additional structural analysis. The ProSA-wed (https://prosa.services.came.sbg.ac.at/prosa.php (accessed on 10 October 2021)) [25,26] was used to assess the quality of the 3D model for PhaC_Ap_.

## 3. Results

### 3.1. Cloning, Expression, and Purification of rPhaC_Ap_

A band of approximately 1131 bp on the agarose gel electrophoregram corresponded to the fragment encoding rPhaC_Ap_ (Figure 1A). The fragment was purified and ligated into the pSol-Tsf vector and successfully verified by colony PCR and DNA sequencing. E. cloni^®^10G was transformed with the recombinant plasmid pSol-Tsf-phaC_Ap_ for the expression of rPhaC_Ap_ with 6x-His-tagged fusion (Figure 1B). Conditions for rPhaC_Ap_ induction were established with the expression strain E. cloni^®^10G. SDS-PAGE showed that rPhaC_Ap_ was present in soluble fraction (SF) after induction with 0.002% (*w*/*v*) l-rhamnose at 30 °C (Figure 2A, lane 3). Increasing the l-rhamnose concentration (0.02–0.2% (*w*/*v*)) induced the amount of soluble rPhaC_Ap_ *where* the maximum expression in SF was observed at 0.2% (*w*/*v*) L-rhamnose (Figure 2A, lane 5). The purified rPhaC_Ap_ from the overexpressing cells induced with 0.2% (*w*/*v*) l-rhamnose showed one distinct band with molecular size of 69 kDa (Figure 2A, lane 6). The presence of N-terminal 6x-His-tagged rPhaC_Ap_ was verified through Western blotting with polyclonal anti-His antibody (data not shown) and polyclonal anti-rPhaC_Ap_ antibody (Figure 2B).

### 3.2. PHA Synthase Activity and Kinetics

The optimal condition for purified rPhaC_Ap_ activity was observed at 30 °C in 100 mM Tris-HCl buffer (pH 8.0) in the presence of 3HB-CoA as substrate. The Michaelis–Menten fitting of the data is shown in Figure 3A. The V_max_, K_m_, and k_cat_ values for 3HB-CoA of the purified rPhaC_Ap_ were 24.5 ± 2 μmol/min/mg, 31.3 ± 2 μM, and 412.7 ± 2 1/s, respectively.

### 3.3. Size-Exclusion Chromatography

Size-exclusion chromatography was performed to examine the multimeric formation of PhaC_Ap_. Two peaks were detected at 19.0 and 24.5 min (Figure 3B). According to the calibration curve generated from the elution times of the molecular standard, the molecular weights corresponding to peaks with elution times of 19.0 and 24.5 min were estimated to be 141 and 71 kDa, respectively, which corresponds to the dimeric rPhaC_Ap_ (138 kDa) and monomeric rPhaC_Ap_ (69 kDa).

### 3.4. Sequence Analysis and Multiple Sequence Alignment for Homology Modeling

The gene for PhaC_Ap_ in *A. platensis* had a length of 1095 base pairs and encoded a protein consisting of 364 amino acid residues with a molecular weight of 42 kDa. Due to the low sequence identity between PhaC_Ap_ and the structurally known PhaC homologs, multiple sequence alignments with intact structural data were made to provide a more reliable basis for the construction of a 3D model for PhaC_Ap_. A structure-based sequence alignment, pre-aligned for PhaC_Cs_ and PhaC_Cn_ structures, was used to align PhaC_Ap_ and four other related PhaC sequences. The resulting alignment is shown in Figure 4. According to the structure-based alignment, PhaC_Cs_ and PhaC_Cn_ share 51.5% sequence identity with each other, and 20% sequence identity with PhaC_Ap_. Table 1 displays the conserved amino acid residues found in PhaC_Ap_, PhaC_Cn_ (Cupriavidus necator PhaC), and PhaC_Cs_ (*Chromobacterium* sp. USM2 PhaC).

Since the experimental analysis revealed that rPhaC_Ap_ forms a catalytically active dimer, we used the asymmetric dimer of the catalytic domain of PhaC_Cs_ (6K3C) as a template for modeling to predict the overall fold and pinpoint the catalytically important residues and structural features of rPhaC_Ap_. Similar to the template structure of PhaC_Cs_, the 3D model for the PhaC_Ap_ is an asymmetric dimer in which the CoA-free monomer is in the closed conformation and the CoA-bound monomer in the open conformation (Figure 5A). The α/β hydrolase fold, consisting of the α/β core subdomains and the CAP subdomains (residues Val175-Pro301), forms the dimer interface in both the closed and open monomers of the PhaC_Ap_ dimer. The α/β core subdomain consists of β-sheets (β1-β8) and α-helices (α1-α5) in both monomers. However, the closed and open conformations represent a distinct conformation, particularly in the CAP domain (Figure 5B). The closed form of the CAP domain is composed of four α-helices (αA-αD) and three 3_10_-helices (ηA-ηC) in the sequence (ηA-αA-αB-αC-αD-ηB-ηC), and make close contacts with the α/β core subdomain. The αB helix, which is made up of 26 amino acid residues (Asp215-Ile240), is a unique feature of the CAP subdomain. The αB helix, αB-αC loop, and αC helix of the CAP subdomain provide protection to the catalytic triad residues (Cys151-His339-Asp310) to a significant extent. On the other hand, the open form of the CAP subdomain comprises five α-helices (αA-αF) and a short 3_10_-helix (ηA-ηF) in the sequence (ηA-αA-αB-αC-αD-ηB-ηF-ηC-ηD). In the open conformation, the fold of the CAP subdomain (residues Val175-Pro301) differs from that of the closed form, and the subdomain does not provide coverage to the catalytic triad residues. The dimer interface is formed by the CAP subdomain of both chains (Figure 5A,B). Analysis of the homology model for PhaC_Ap_ with the ProSA-web gave a Z-score of −6.24 (Figure 6A), and the crystal structure of PhaC_Cs_ gave a Z-score of −8.1 (Figure 6B). These Z-scores are within the range of the typical scores for experimentally determined structures of a similar size. Furthermore, the predicted secondary structure for PhaC_Ap_ matches well with the secondary structure elements in the PhaC_Cs_ and PhaC_Cn_ crystal structures. Hence, the overall fold of PhaC_Ap_ could be modeled with relatively good reliability.

### 3.5. The Active Site and Catalytic Mechanism of PhaC_Ap_

The overall architecture of the active site in PhaC_Ap_ is very similar to that of PhaC_Cs_. In the closed monomer of PhaC_Cs_, the αB-αC loop (Leu380-Lys382) in the CAP subdomain blocks the catalytic triad. In the closed-form monomer of rPhaC_Ap_, the catalytic triad is blocked by residues Met241-Ser243 of the αB-αC loop in the CAP subdomain (Figure 7A,B). Residues Met241-Ser243 of the αC-αD loop in the CAP subdomain have a similar role in the open conformation of PhaC_Ap_ (Figure 7C,D). In PhaC_Ap_, the open conformation exposes the catalytic Cys-His-Asp triad (Cys151-His339-Asp310). Residues Asp86, Leu87, His311, and His339 near the CoA binding site are conserved in PhaC_Ap_, PhaC_Cs_, and PhaC_Cn_ (Table 1). The PhaC_Ap_ catalytic triad (Cys-Asp-His) cleft is situated between the CAP subdomain and the α/β core subdomains. The β5-α3 loop containing Cys151, the β7-α4 loop containing Asp310, and the β8-α5 loop containing His339 are the components that make up the α/β core subdomains. The extended pantetheine arm of CoA attaches to the active site, which is close to the catalytic triad, and the ADP moiety of CoA is positioned there. Cysteamine containing the terminal thiol (SH) group is close to Cys151 (Figure 7C). The ribose ring is shielded by the hydrophilic residue Glu37 of the extended N-terminal loop, while the Val85 residue from β3-α1 loop interacts with the adenine moiety of CoA. The interaction with CoA is facilitated by Met241-Ser243, which is located in the αC-αD loop of the CAP subdomain. The hydrophobic nature of the cleft is due to the presence of nonpolar residues from the α/β core subdomains, namely Asp86, Val85, Leu312, and Val313. The side chain of Asp310 forms a hydrogen bond with the pantetheine arm of CoA. In addition, the αC-αD loop of the enzyme locks the pantetheine arm of CoA, as shown in Figure 7C, through the side chain carbonyl groups of Leu312 and Val313.

## 4. Discussion

In this study, we cloned class III PhaC from *A. platensis* (rPhaC_Ap_) and produced rPhaC_Ap_ with hydrophilic His-tagged fusion using a bacterial protein expression system. We demonstrated that His-tagged rPhaC_Ap_ was highly induced with 0.2% (*w*/*v*) l-rhamnose at 30 °C. The purified His-tagged rPhaC_Ap_, which represented a molecular size of 69 kDa, was used directly as an antigen for immunization without further depleting His-tag [27]. The presence of N-terminal His-tagged rPhaC_Ap_ was verified through Western blotting with polyclonal anti-rPhaC_Ap_ antibodies. The lower faint band was the minor protein contaminants. These minor proteins showed no changes without or with induction by increasing concentration of L-rhamnose (Figure 2A, lanes 2, 3, 4, and 5). In addition, size-exclusion chromatography revealed the active dimerization of rPhaC_Ap_ subunit (138 kDa). These results suggested that the PhaC_Ap_ exists as a dimeric assembly in solution. On the basis of previous observations, PhaC synthase likely forms either various dimers or multimers other than monomers [28,29,30]. Furthermore, a dimeric form of PhaC synthase was important in its catalytic function, although, in general, the monomeric and dimeric forms of PhaC exist in equilibrium [31,32,33]. The Michaelis–Menten fitting of the data shown in Figure 3A demonstrated that *K_m_* and *k_cat_* values for 3HB-CoA of the purified rPhaC_Ap_ were 31.3 ± 2 μM and 412.7 ± 2 1/s, respectively. Numerous studies have reported kinetic parameters of PhaC. One of those demonstrated that the *K_m_* value for 3HB-CoA of PhaCE from *Synechocystis* sp. PCC 6803 (rPhaCE) was 478 ± 31 µM [20], whereas the *K_m_* value of PhaC from *Aeromonas caviae* FA440 (rPhaC_Ac_) was 77 ± 5 µM [34]. In addition, *K_m_* and *k_cat_* values of the purified rPhaC_Av_ (*Allochromatium vinosum*) were 0.11 mM and 508 1/s, respectively [35], while those of the purified rPhaEC_Es_ (*Ectothiorhodospira shaposhnikovii*) were 0.065 mM and 320 1/s, respectively [36], and 0.13 mM and 3920 1/min, respectively for rPhaEC_Av_ (*Allochromatium vinosum* [37]. Probably, high values of *K_m_* correspond to the structure of PHA synthase. The rPhaCE has a lower enzyme affinity for substrates than rPhaC. The catalytic activity of PHA synthase is easily influenced by abiotic factors, i.e., pH, ionic strength, or temperature [38]. Although this enzyme has frequently been tested for activity using both Tris-HCl and phosphate-buffered solutions, at the same concentration and pH, Tris-HCl buffer exhibits primarily greater enzyme activity [38]. In the present study, the highest rPhaC_Ap_ activity was observed in 100 mM Tris-HCl buffer (pH 8.0) at 30 °C. Additionally, the PHA synthase activity from *Ralstonia eutropha* was reduced when the concentration of the buffer solution increased [38]. This implies that ionic strength affects PHA synthase activity. Additionally, the cultivation temperature could significantly influence PHB production. In terms of the class III PHA synthase, *E. coli* JM109 harboring the *phaECAB* expression plasmid was found to be capable of producing PHB. The maximum PHB production (0.64 g/L) was found at 30 °C among the four temperatures tested (25 °C, 30 °C, 37 °C, and 42 °C), indicating their differences in catalytic activity [39]. In addition, when BSA (0.2 mg/mL) was added in the assay mixture, the PHA synthase activity from *Allochromatium vinosum* (PhaEC_Av_) was increased [35]. BSA has been shown to bind PHB granules in vitro, and it was proposed that BSA interacts with hydrophobic polymers to prevent the increasing chain from obstructing the active site, which led to the observed increase in activity. In the present study, inclusion of BSA (0.2 mg/mL) in the reaction mixture somehow enhanced the activity of rPhaC_Ap_. Moreover, it was reported that a *R. eutropha* PHA synthase mutant with a high catalytic activity can synthesize higher molecular weight P (3HB) compared with wild type or mutants with lower catalytic activities, indicating that the catalytic activity of PHA synthase affects the molecular weight of PHA [40].

The PhaC_Ap_ forms are very similar to those of PhaC_Cs_. The αB-αC loop undergoes conformational changes in the PhaC_Cs_ crystal structure, causing the C-terminal end region of the α/β core subdomains to become disordered. As a result, the αB and αC helices move away from the active site, allowing the CoA molecule to enter the active site cleft. [9]. The movement of the αB helix and αC helix of the CAP subdomain in both PhaC_Cs_ and PhaC_Ap_ is similar to a retracting “Boom gate” mechanism, which covers the catalytic Cys-His-Asp triad and blocks substrate entry [9]. In the open conformation of PhaCCs, the αC-αD loop in the CAP subdomain contains Leu380-Lys382 residues that form polar interactions with CoA, which is critical for PhaC activity. Additionally, the CoA moiety of acyl-CoA substrates plays an important role in this process. Zhang et al. [41] reported that the Class III synthase (PhaC-PhaE) from *Allochromatium vinosum* (PhaCE_Av_) appears to require the CoA molecule for binding to its substrate. These results are in agreement with the hypothesis that the complex between PhaC and the substrate acyl-CoA is formed by the interaction between PhaC and the CoA moiety of the substrate. The CAP domain may play a significant role in the catalytic activity of PHA synthase, because local conformational changes in the α/β core subdomain are caused by conformational changes of the CAP subdomain. The activity of PHA synthase is reliant on the presence of the catalytic triad C-H-D, which refers to the amino acid residues cysteine, histidine, and aspartate. Interestingly, this same triad is also responsible for the catalysis of lipases, but with the substitution of serine in place of cysteine (S-H-D) [8]. In the crystal structure of PhaC_Cs_, the Cys291, Asp447, and His477 that form the catalytic triad are crucial for its catalytic activity [9]. The most striking difference between the crystal structures of PhaC*_Cs_* and PhaC*_Cn_* compared to PhaC*_Ap_* is the lack of so-called protruding structure regions (PS-region in Figure 4), which are also lacking in some other PhaC proteins [42].

The catalytic triad in PhaC_Cs_ has been proposed to function in PHA biosynthesis in the following way. First, the 3HB-CoA molecule enters the active site tunnel of PhaC, where His477 facilitates deprotonation of the thiol group of the catalytic triad residue Cys291. This results in the formation of a covalent bond between 3HB and Cys291, and the release of CoA from the tunnel. Asp447 plays a role in the PHA elongation process by participating in the attack on the acyl-CoA substrate, specifically targeting the hydroxyl group within the acyl moiety. After the release of CoA, the newly attacked 3HB-CoA enters the active site and the cycle repeats. In this way, the polymer chain is elongated, resulting in the formation of (3HB)_n+1_ covalently bound to the Cys residue. Therefore, at the end of each cycle, the 3HB polymer is bound to the enzyme [43]. In addition, in the CoA-bound monomer of the PhaC_Cs_ dimer, the replacement of aspartic acid with asparagine (D447N) forms hydrogen bonds with the carbonyl group of CoA [14,44], and it was reported that a D480N mutation in 6x-His-tagged PhaC_Cn_ reduced the reaction rate to the 0.0008 units/mg from 20 unit/mg observed in wild type. Furthermore, the mutation of Phe318 in the pocket in *Ralstonia eutropha* PHA synthase (PhaC_Re_) led to a decrease in synthase activity [42]. Harada et al. [28] demonstrated that mutation of Phe318 in PhaC_Ac_ resulted in a significant reduction in polymer synthesis. Moreover, Müh et al. [45] showed that PhaC from *Chromatium Vinosum* was capable of in vitro PHA polymerization in the absence of PhaE, and PhaC alone was much more susceptible to such inhibition of PMSF in comparison with PhaEC.

## 5. Conclusions

In this study, the recombinant PhaC_Ap_ enzyme was constructed and produced in an *E. cloni^®^*10G system. Our homology model shows that Cys151, Asp310, and His339 are the important catalytic triad residues that determine the pocket size of PhaC_Ap_. Therefore, it is reasonable to hypothesize that a mutation at these positions would have a significant influence on the pocket depth and PhaC synthase activity. To better understand the regulation of PHA synthesis in *A. platensis*, the kinetics of enzymes with catalytic triad (Cys-Asp-His) mutations would be valuable for further investigation. The proper orientation of the substrate may increase the efficiency of the catalytic reaction. Additionally, the model for an asymmetric PhaC_Ap_ homodimer provided insight into the putative mechanism and function of PHA synthase in *A. platensis*.

## Figures and Tables

**Figure 1 biology-12-00751-f001:**
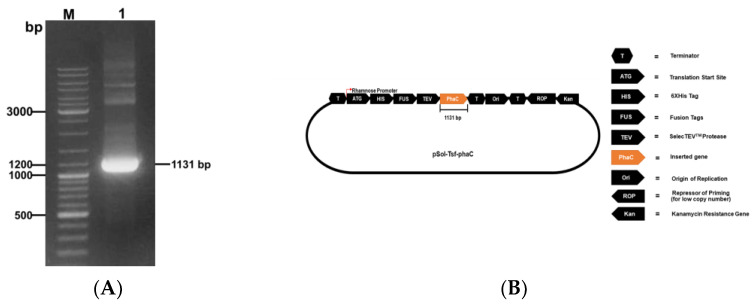
(**A**) PCR amplification from *A. platensis* DNA, lane: M marker; lane 1: PCR fragment of *A. platensis* phaC (1131 bp) (**B**) Schematic map of pSol-Tsf-phaC_Ap_ plasmid carrying *A. platensis* phaC gene).

**Figure 2 biology-12-00751-f002:**
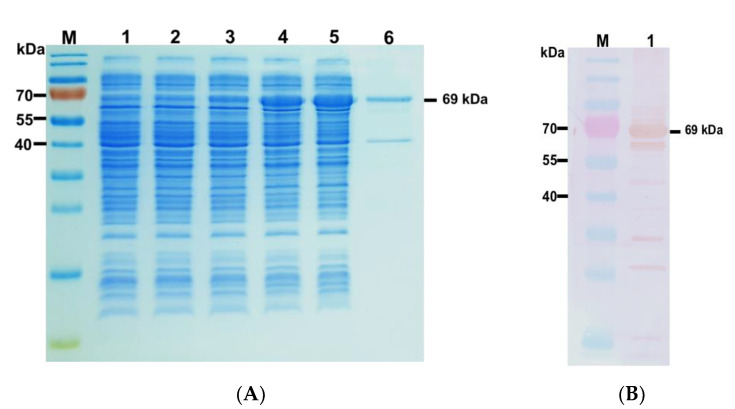
(**A**) SDS-PAGE representing rPhaC_Ap_ protein expression in E. cloni^®^10G. The target protein is about 69 kDa. Lane M: protein marker (in kDa); lane 1: crude protein fraction; lane 2: soluble protein fraction (SF) before l-rhamnose induction; lane 3: SF after 0.002% (*w*/*v*) l-rhamnose induction; lane 4: SF after 0.02% (*w*/*v*) L-rhamnose induction; lane 5: SF after 0.2% (*w*/*v*) L-rhamnose induction; lane 6: purified rPhaC_Ap_ obtained from SF after 0.2% (*w*/*v*) l-rhamnose induction. (**B**) Western blot detection of rPhaC_Ap_ protein with polyclonal anti-rPhaC_Ap_ antibody which corresponds to protein band of approximately 69 kDa. Lane M: protein marker (in kDa); lane 1: purified soluble protein fraction containing rPhaC_Ap_ protein after 0.2% (*w*/*v*) L-rhamnose induction.

**Figure 3 biology-12-00751-f003:**
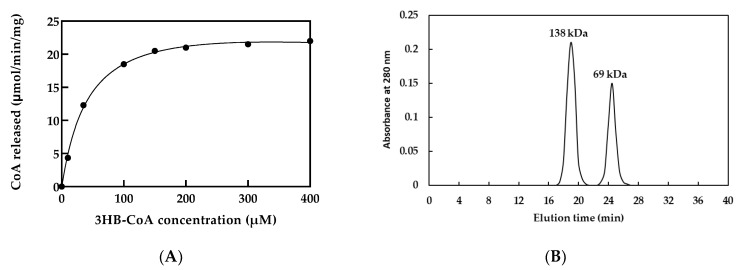
(**A**) The Michaelis–Menten plot of purified rPhaC_Ap_ (**B**) Size-exclusion chromatography of rPhaC_Ap_.

**Figure 4 biology-12-00751-f004:**
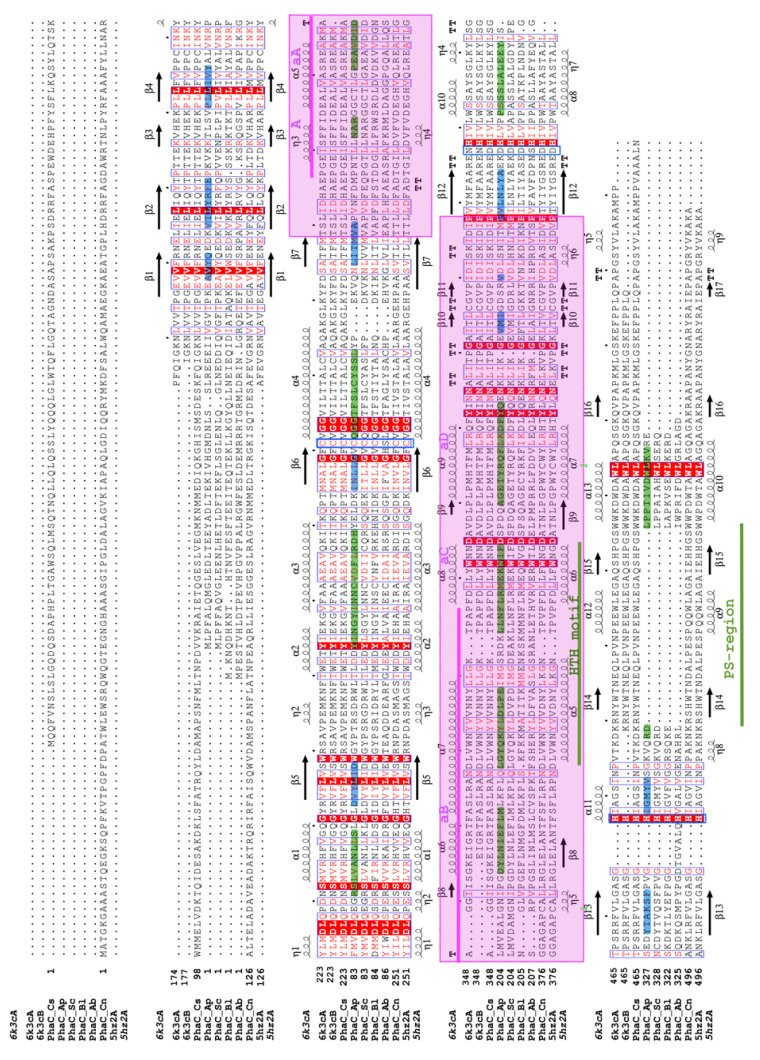
Multiple sequence alignment used for modeling. The PhaC proteins of PhaC_Ap_, PhaC_Ss_, PhaC_Pl_, and PhaC_Ab_ were aligned to the pre-aligned structure-based alignment of *Chromobacterium* sp. USM2 PhaC_Cs_ (PDB ID 6K3C) and Cupriavidus necator PhaC_Cn_ (PDB ID 5HZ2). The secondary structures of PhaC_Cs_ and PhaC_Cn_ are shown above and below the alignment, respectively. The predicted secondary structure of PhaC_Ap_ is highlighted, with helices colored green and beta-strands colored blue. Residues that are conserved are depicted with a red background. The catalytic triad residues (Cys-His-Asp) are marked with blue boxes. CAP subdomains are highlighted in pink and the Lid area with a pink line. The HTH motif and lacking PS-region are shown with a green line and marked.

**Figure 5 biology-12-00751-f005:**
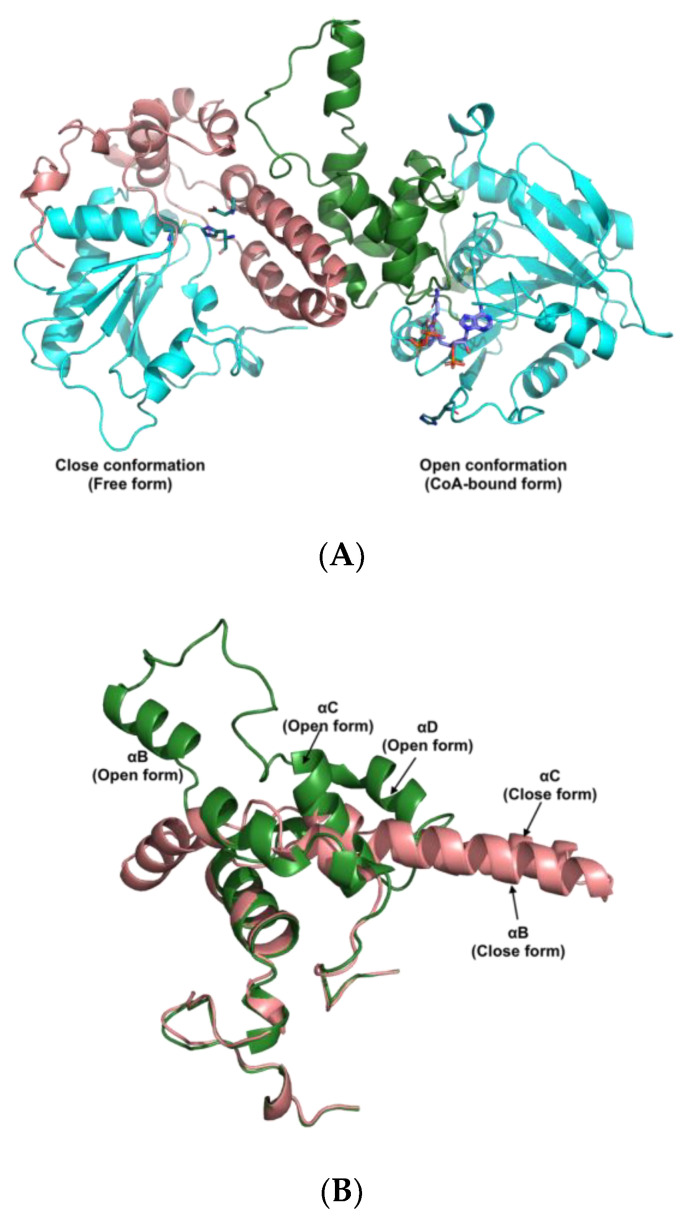
(**A**) Model for the PhaC_Ap_ homodimer consisting of free form and CoA-bound form. The α-helical CAP subdomains in the dimer are colored in salmon and forest (residues 175-301). The catalytic triad (Cys151-His339-Asp310) is represented by a stick in deep teal color. The CoA is a purple stick. (**B**) Comparison between CAP subdomains of free form (salmon) and CoA-bound form (forest).

**Figure 6 biology-12-00751-f006:**
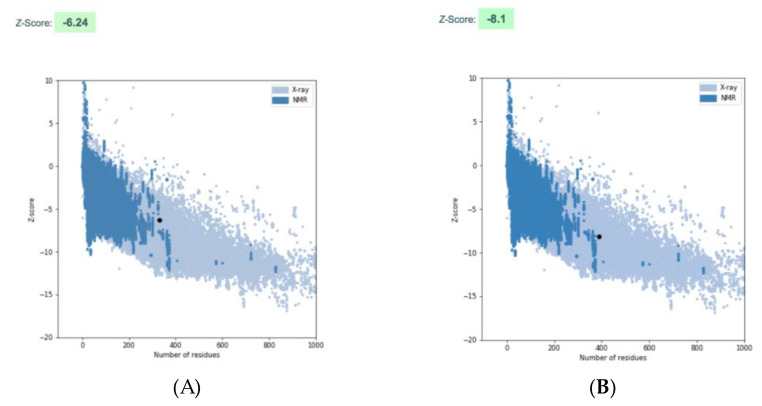
Quality assessment of the PhaC_Ap_. The graphs reveal the quality assessment of the model structures using ProSA-web [25,26]. (**A**) The ProSA-web scores (black dot) for the 3D model of PhaC_Ap_ are based on the PhaC_Cs_. (**B**) The ProSA-web scores (black dot) for the 3D model of PhaC_Cs_.

**Figure 7 biology-12-00751-f007:**
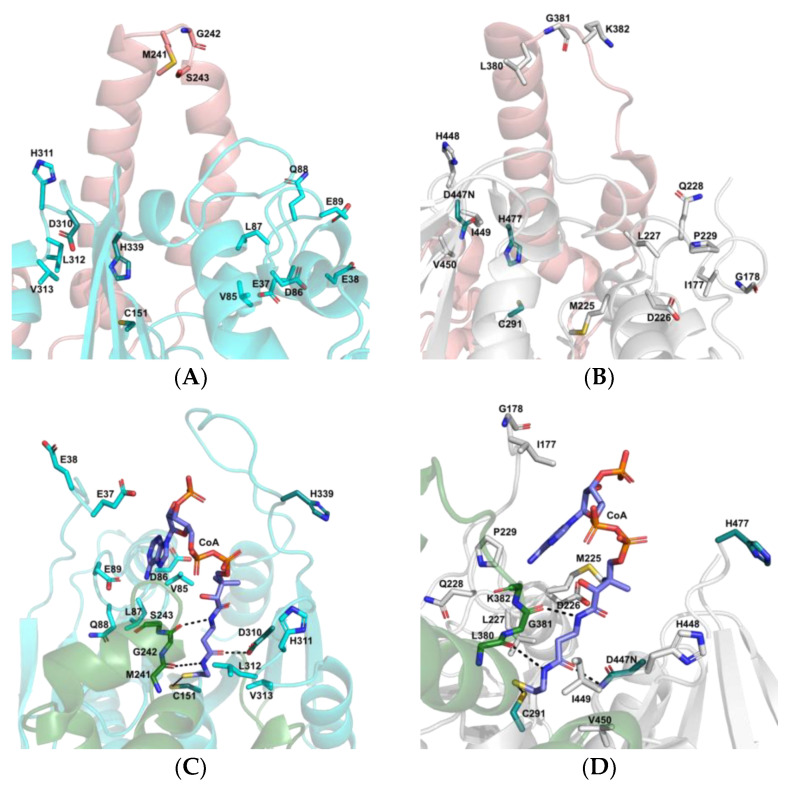
Side view of the closed and open active site cleft. (**A**,**C**) The active site of PhaC_Ap_ with the catalytic triad Cys151-His339-Asp310 in the closed and open conformation. (**B**,**D**) The active site of PhaC_Cs_ (6K3C) in the closed and open conformation. The catalytic triad residues displayed as deep-teal sticks and the CoA as purple sticks. The residues of the CAP subdomain from the closed and open conformation are presented in salmon and forest color, respectively.

**Table 1 biology-12-00751-t001:** List of the conserved amino acid residues in PhaC_Cn_, PhaC_Cs_, and PhaC_Ap_. The catalytic triad residues are depicted with a blue background.

Conserved Residue	PhaC_Cn_	PhaC_Cs_	PhaC_Ap_	Conserved Residue	PhaC_Cn_	PhaC_Cs_	PhaC_Ap_
			(This Study)				(This Study)
V	V211	V183	V43	L	L316	L288	L148
T	T212	T184	T44	G	G317	G289	G149
V	V217	V189	V49	C	C319	C291	C151
L	L225	L197	L57	G	G321	G293	G153
P	P230	P202	P62	G	G322	G294	G154
K	K234	K206	K66	Y	Y406	Y373	Y234
P	P239	P211	P71	G	G414	G381	G242
L	L240	L212	L72	W	W425	W392	W256
L	L241	L213	L73	D	D428	D395	D259
V	V243	V215	V75	L	L441	L408	L272
N	N248	N220	N80	Y	Y445	Y412	Y276
D	D254	D226	D86	N	N448	N415	N279
L	L255	L227	L87	L	L450	L417	L280
Q	Q256	Q228	Q88	G	G454	G421	G284
S	S260	S232	S92	D	D464	D431	D294
V	V262	V234	V94	I	I468	I435	I298
G	G269	G241	G101	P	P471	P438	P301
V	V272	V244	V104	D	D480	D447N	D310
L	L274	L246	L106	H	H481	H448	H311
W	W277	W249	W109	V	V483	V450	V313
T	T288	T260	T120	G	G507	G476	G338
Y	Y292	Y264	Y124	H	H508	H477	H339
I	I293	I265	I125	W	W554	W523	W359
N	N314	N286	N146	L	L555	L524	L360

## Data Availability

All data obtained during this work is available from the authors on request.

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
