# Peer review of "Characterization and Homology Modeling of Catalytically Active Recombinant PhaCAp Protein from Arthrospira platensis"

_biology, 2023, doi:10.3390/biology12050751_

Round 1

Reviewer 1 Report

The authors gave a very detailed and thorough story about the enzymatic activity of PHaCap. The experimental and discussion is very solid. However, I'd like to make the following suggestions:

1. It is helpful to add a figure for the introduction, especially drawing the biosynthesis pathway of Polyhydroxybutyrate.

2. The Kcat and Km were measured using CoA release. This may be due to 3HB-CoA hydrolysis. It is better to show the direct production of Polyhydroxybutyrate.

Reviewer 2 Report

In this study, Class III PHA synthase (PhaCAp) from the cyanobacterium Arthrospira platensis was characterized. The Vmax, Km, and kcat values for R-3-hydroxybutyryl coenzyme A (3HB-CoA) of the purified rPhaCAp were 4.5±2 32 umol/min/mg, 31.3±2 μM, and 412.7±2 1/s, respectively. Size exclusion chromatography revealed that rPhaCAp exists as an active dimer. The overall fold was predicted using the 3D structural model for rPhaCAp. This paper covers all the analyzes for PhaC in general and provides useful information as a new PhaC.

This PHA synthase seems to show sufficient polymerization activity without the need for a PhaE subunit. Why was it classified as Class III?

The size of the PhaCAp subunit is described as 69 kDa, it is the typical size of a Class I PhaC.

Size exclusion chromatography revealed that rPhaCAp exists as an active dimer but still presents monomer PhaC as shown in Fig. 3B. Does the monomer PhaC have polymerization activity? Did you exclude the monomer PhaC when calculating kcat value?
